spectroscopy/environmental chemistry/biomaterials

FTIR, UV-Vis, eggshell, insecticide, adsorption

**Author for correspondence:**
Neslihan Kaya Kınaytürk
e-mail: nkinayturk@mehmetakif.edu.tr

This article has been edited by the Royal Society of Chemistry, including the commissioning, peer review process and editorial aspects up to the point of acceptance.

# Eggshell as a biomaterial can have a sorption capability on its surface: A spectroscopic research

Neslihan Kaya Kınaytürk[1], Belgin Tunalı[1] and
Deniz Türköz Altuğ[2]

[1]Faculty of Arts and Sciences, Department of Nanoscience and Nanotechnology, Burdur Mehmet Akif Ersoy University, Burdur, Turkey
[2]Isparta Vocational School of Health Services, Süleyman Demirel University, East Campus, Isparta 32260, Turkey

 NKK, 0000-0002-2170-1223; BT, 0000-0003-0768-679X;
DTA, 0000-0002-1861-6263

In this study, eggshell as a biomaterial was used as an adsorbent. This natural waste material is easy to access and cost-free. The surface of the eggshell with its porous structure showed affinity to adsorb damaging chemicals. In particular insecticides cause serious environmental pollution in agriculture, and this is a general problem all over the world. The aim was to remove insecticides from the environment and monitor the pesticides on the surface of eggshells by atomic force microscopy (AFM) images, Fourier transform infrared (FTIR) and UV/Vis spectroscopic techniques. Five types of eggshells, *Denizli Hen*, *Coturnix Coturnix Japonica*, *Light Brahma Chicken*, *Alectoris Chukar* and *ISA Tinted -White*, were used. Since they are commonly used, Cypermethrin, Deltamethrin and Indoxacarb were chosen as insecticide samples. The interaction effect of insecticides on the surface of eggshells was determined by AFM images; it was seen that the semispherical surface structures of the eggshells were flattened after adsorption. FTIR spectroscopy was used both to detect structural analysis and to determine the adsorption influence. In addition, UV-Vis spectroscopy was performed to evaluate the adsorption and desorption process. Porous media of different types of eggshells with an aqueous solution of insecticides had an electronegativity attractive surface which makes it an ideal adsorbent via hydroxyl groups.

# 1. Introduction

Egg is a food product that is used in large quantities by food manufacturers, restaurants and households, and eggshells are

disposed of as solid waste. Today, waste materials are considered by everyone to be important because of they are natural and can contribute to the economy.

Due to the high amount of waste, various research has been conducted to determine useful application areas. Since it is a source of organic P, it is frequently used as a natural fertilizer in plant breeding [1]. Furthermore, the fact that each eggshell contains between 7000 and 17 000 pores makes it attractive to use as an adsorbent [2]. Eggshell is a natural material, more than 95% of its content contains calcium carbonate ($CaCO_3$) and it has been the subject of many interesting studies due to its porous structure. The use of natural materials like eggshell to protect the ecosystem from pollutants has been a subject of great interest for years. While it is used in nanocoating [3,4], on the one hand, it is also used in the pharmaceutical industry due to its strengthening effects on bone, tooth and cartilage structure after some synthesis work due to the percentage excess of $CaCO_3$ content [5–9]. On the other hand, removal of some toxic chemicals has also been investigated through some studies [10–12]. Besides this, there are some interesting studies related to biodiesel [13,14]. There are many research articles on the removal of heavy metals from waste water. The porous structure of the eggshells has also been used to remove Pb, Cu, Cd [15,16], Hg ions and methyl violet [17] and unwanted substances such as Congo red [18] and oil [19] from aqueous solutions.

In this study, five different types of eggshells (*Denizli Hen* (*DH*), *Light Brahma Chicken* (*LBC*), *ISA-Tinted-White* (*ITW*), *Alectoris Chukar* (*AC*) and *Coturnix Coturnix Japonica* (*CCJ*)) were used. Synthetic insecticides against insects, which are widely used in households and in agriculture, are particularly harmful to humans, animals, water and soil because of their low solubility in prepared mixture and difficulty of being removed from the environment [20–24]. Synthetic insecticides are mobile in the soil [20]. Therefore, it is necessary to prevent their spread from the soil to the environment. In the literature, it is possible to find many studies on the removal of insecticides from the environment [22,23,25]. In this study, Cypermethrin (CM), Deltamethrin (DM) and Indoksacarb (IC) insecticides, which are widely used in agriculture, were selected. CM [20,26] and DM [27] are defined as synthetic pyrethroids and IC [28] is defined as oxadiazine insecticide. The aim was to remove the commonly used insecticides from the environment via eggshells.

# 2. Material and methods

## 2.1. Material and chemicals

Eggshell samples (*DH, LBC, ITW, AC* and *CCJ*) were obtained from Isparta University of Applied Sciences, Education Research and Application Farm. Eggshell samples were taken from incubation wastes of this farm. In addition, insecticides (CM, DM and IC) were provided by Hektas T.A.Ş., Agrobest Group T.A.Ş. and Koruma Klor Alkali T.A.Ş., respectively. They are normally used for commercial purposes in Turkey. In this work, insecticides were used without purification. Chemical information on these compounds is shown in table 1. Photos of poultry and their eggs which we used are shown in figure 1.

## 2.2. Experimental

For cleaning of the eggshells, all types (*DH, LBC, ITW, AC* and *CCJ*) were washed with tap water and then boiled at 200°C for about 2 h, and the membrane of eggshells was peeled off. Then they were boiled again with distilled water for 1 h and filtered. Finally, they were left in the ultrasonic bath for 30 min with acetone to remove any residual materials. Then, the eggshells were dried on coarse filter paper at ambient temperature. The Fourier transform infrared (FTIR) spectroscopic investigation (grounded) and atomic force microscopy (AFM) analysis (without powder form) were performed for the characterization of eggshells before the adsorption process. After the adsorption process, FTIR, Ultra Violet-Visible (UV-Vis) spectroscopic and AFM analyses were used.

## 2.3. Characterization studies

### 2.3.1. For AFM analysis

Eggshell samples with an average surface area of 0.25 cm$^2$ were analysed before and after adsorption. All AFM images were acquired in non-contact mode with an image resolution of $250 \times 250$ pixels.

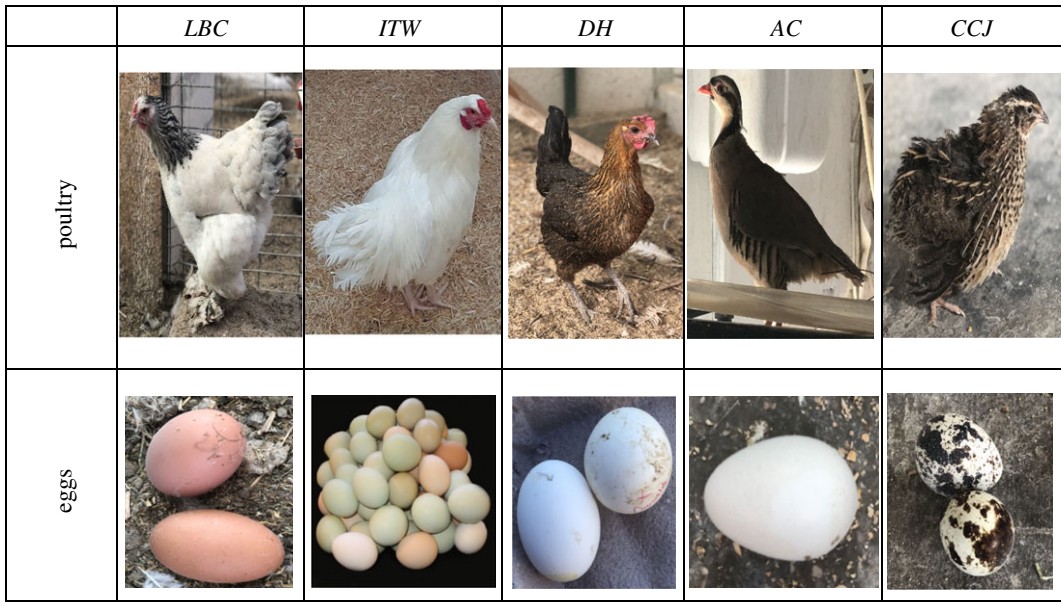

**Figure 1.** Photos of poultry and their eggs.

**Table 1.** Chemical information of used insecticides.

| compounds | structure | chemical formula | chemical name |
|---|---|---|---|
| CM | | $C_{22}H_{19}Cl_2NO_3$ | 3-(2,2-dichlorovinyl)-2,2-dimethylcyclopropanecarboxylic acid and the alcoholic hydroxy group of hydroxy(3-phenoxyphenyl) acetonitrile [26,27]. |
| DM | | $C_{22}H_{19}Br_2NO_3$ | (S)-alpha-cyano-3-phenoxybenzyl (1R)-cis-3-(2,2-dibromovinyl)-2,2-dimethylcyclopropan carboxylate [27]. |
| IC | | $C_{22}H_{17}ClF_3N_3O_7$ | Methyl 7-chloro-2,5-dihydro-2-[N-(methoxycarbonyl)-4-(trifluoromethoxy) anilinocarbonyl]indeno[1,2-e][1,3,4]oxadiazine-4a(3H)carboxylate [28]. |

### 2.3.2. For FTIR spectroscopic analysis

Eggshells with an average 0.1 g were grounded. FTIR were used to record the spectra of the samples in a range of 4000–400 cm$^{-1}$, using the KBr pellet technique. FTIR analyses of pure CM, DM and IC insecticides were performed using the ATR technique in the same range.

## 2.4. Theoretical analysis of electronegativity

For insecticides, Gaussian 09 [29] software package was used to calculate the value of electronegativity. The calculations of HOMO-LUMO energy values were performed by applying the time-dependent DFT (TD-DFT) method with the B3LYP function and the 6–311++ G(d, p) basis set.

### 2.4.1. Adsorption studies

#### 2.4.1.1. For FTIR spectroscopic analysis

The mixtures were prepared with 25 ml insecticide and 75 ml distilled water. (ratio 1/3 = insecticide/ distilled water); 1.5 g powdered eggshells (*DH, LBC, ITW, AC* and *CCJ*) were treated with a 15 ml mixture. They were kept for 48 h and then filtered. Finally, FTIR analysis was performed.

#### 2.4.1.2. For UV-Vis spectroscopic analysis

Powdered 0.15 g eggshells were treated with 15 ml of 250 ppm mixture. UV-Vis spectroscopic measurements were taken by diluting to 5 ppm in 10 different time intervals (15th min, 2th h, 4th h, 6th h, 8th h, 12th h, 24th h, 28th h, 32th h and 48th h).

### 2.4.2. Desorption studies

After two days of adsorption study, the samples were filtered and dried. Each 0.01 g sample was weighed and 10 ml of distilled water was added. Then, samples were measured by UV-Vis spectrometer at seven different time intervals for 24 h.

## 3. Results and discussion

### 3.1. AFM analysis

AFM results were used to investigate the topological information of the eggshells and also their effectiveness in obtaining surface structure in detail.

Figure 2 shows a series of AFM topographical two- and three-dimensional images of the pure and adsorbed form eggshells. Figure 2*a* shows the pure *DH* eggshell view. According to this figure, DH eggshells have a partially channeled appearance. Figure 2*b* is the view of the *DH* eggshell with an adsorbed form of CM. According to this image, more channels are closed and flattened. Figure 2*c* shows that pure *CCJ* eggshell has a very semi-spherical appearance. In this image, there are lots of small pores distributed all over the surface, and it is more homogeneous compared to figure 2*a*. In figure 2*d*, for the adsorption of DM on *CCJ* eggshell the image shows that eggshell is filled with DM and causes a more flat appearance. According to the AFM, two- and three-dimensional images in *DH* eggshells, the gaps between the halls were partially filled with CM after adsorption. These results are supported by FTIR and UV-Vis spectroscopic analysis.

### 3.2. FTIR spectroscopic analysis

We used spectroscopy for characterizing chemical bonds on the surface of the adsorbant. Variations in functional groups give clues about adsorption on eggshell samples. Detailed FTIR assignments of pure eggshells are presented in table 2. Six characteristic vibration bands were observed in all eggshells used in this study before adsorption; these bands were around 2516, 1799, 1424, 1083, 874 and 712 cm$^{-1}$. These vibration bands were also observed in some studies before [31,34,35].

The bands observed around 874 and 712 cm$^{-1}$ are assigned to the out-of-plane and in-plane deformation vibration bands, respectively, of $CaCO_3$ [33,34]. Tatzber *et al.* observed these vibration bands at 871 and 712 cm$^{-1}$ [30]. The most prominent characteristic band of eggshells is the C-O stretching vibration band belonging to carbonate. Eleta *et al.* observed this band at 1436, Yusuff *et al.* observed it at around 1424 cm$^{-1}$, and it was observed at 1424 cm$^{-1}$ in this study [11,32]. Similarly, the weak band around 1799 cm$^{-1}$ corresponds to C=O bonds related to carbonate and a shoulder appears at 1084 cm$^{-1}$ which is attributed to the symmetric stretching of $CO_3$ [35,36]. Tatzber *et al.* assigned the 2506 cm$^{-1}$ band as an important carbonate vibration band. This band was observed at 2516 cm in this study [30].

FTIR assignments of pure insecticides and forms adsorbed on eggshells are presented in table 3, and their FTIR spectra are presented in figure 3. When table 3 is examined, it is concluded that some characteristic bands of insecticides demonstrated their presence on the eggshells. Shifts in the characteristic vibration bands of eggshells used as adsorbent material were observed after the

| | two-dimensional | three-dimensional |
|---|---|---|
| (a) | | |
| (b) | | |
| (c) | | |
| (d) | | |

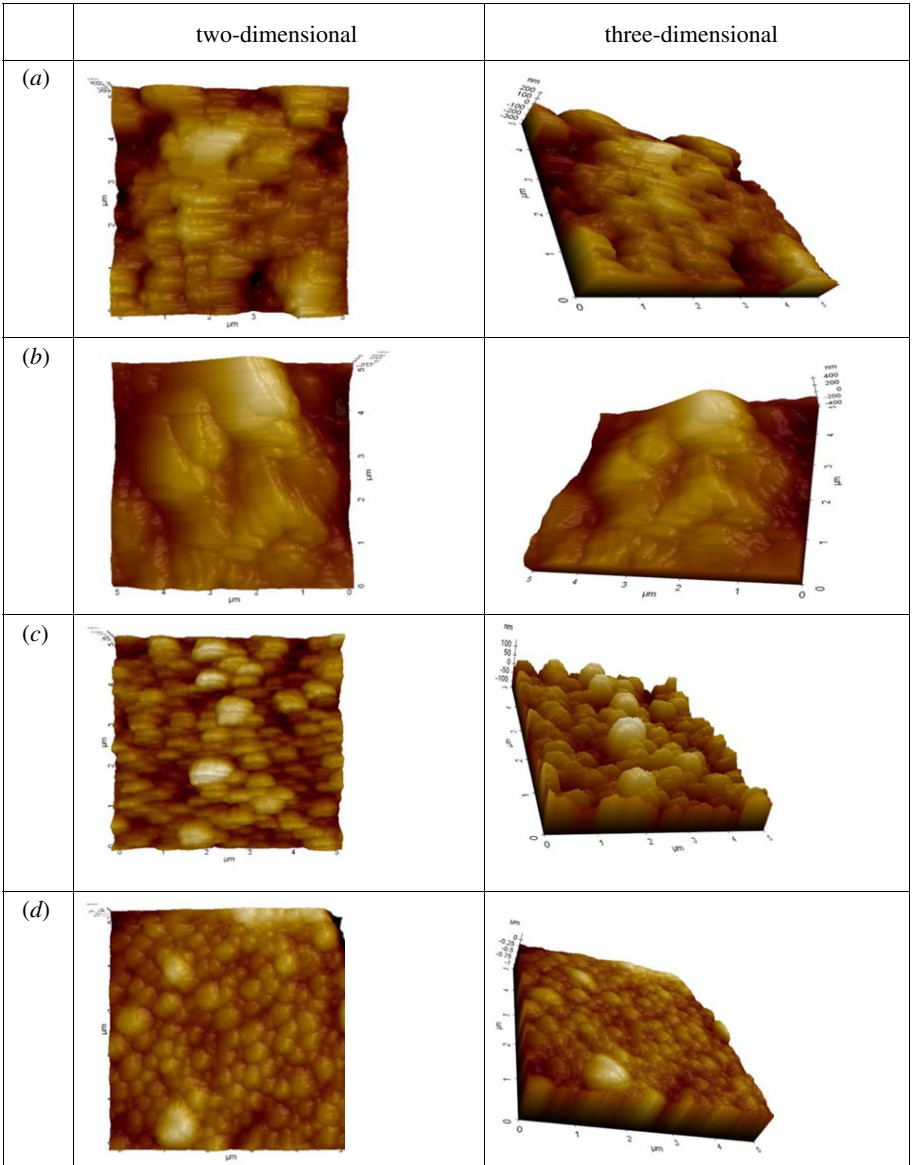

**Figure 2.** Two- and three-dimensional images of the sample (a) DH eggshell, (b) after adsorption of CM into DH eggshell, (c) CCJ eggshell and (d) after the adsorption of DM into CCJ eggshell.

adsorption process. These shifts indicate an interaction between insecticides and eggshells. It can be inferred that some shifts in FTIR assignments are due to the Fermi Resonance in table 3 [37].

In organic molecules, the C-H stretching band is generally observed around 3000–2840 cm$^{-1}$, and these bands are among the most stable bands in terms of positions in the spectrum [38]. In this study, C-H vibration bands were observed in the same region for DM, CM and all adsorbed forms of all eggshells.

The C ≡ N vibration band is characteristic for CM and DM. This band was observed as a very weak band at 2135 cm$^{-1}$ in all samples after adsorption [39]. The C = O vibration band exhibits a very strong IR band in the region between 1850 and 1650 cm$^{-1}$, and this band is related to the stretching vibration of benzene skeleton [40,41]. After the adsorption process, this band appeared at around 1740 cm$^{-1}$. The band at 1256 cm$^{-1}$ is assigned to the C-O-C asymmetrical vibration band [25]. These vibration bands, which are characteristic for DM, CM and IC, were observed around 1250 cm$^{-1}$ in all samples after adsorption. Similarly, the band observed around 1155 cm$^{-1}$ is assigned as a symmetric stretching band of the C-O-C [25]. This band was observed around 1165 cm$^{-1}$ in eggshells adsorbed with DM and IC.

Generally, the C = C stretching vibration band is seen in aromatic compounds form in the region of 1650–1430 cm$^{-1}$. This band was observed in CM and DM adsorbed eggshells and pure CM, DM

**Table 2.** FTIR assignments of pure *DH*, *LBC*, *ITW*, *AC* and *CCJ* eggshells (cm$^{-1}$).

| assignment | DH | LBC | ITW | AC | CCJ | references |
|---|---|---|---|---|---|---|
| $CO_3^{2-}$ stretching | 2516 w | 2515 w | 2515 w | 2516 w | 2516 w | [30] |
| $CO_3^{2-}$ stretching | 1799 w | 1799 w | 1799 w | 1799 w | 1799 w | [30–32] |
| $CO_3^{2-}$ stretching | 1420 s | 1423 s | 1423 s | 1423 s | 1424 s | [30–33] |
| $CO_3^{2-}$ stretching | 1084 sh | 1083 sh | 1082 sh | 1081 sh | 1082 sh | [30,31] |
| $CO_3^{2-}$ in plane deformation | 875 m | 874 m | 875 m | 875 m | 875 m | [30,31,33,34] |
| $CO_3^{2-}$ out of plane deformation | 712 m | 713 m | 712 m | 713 m | 713 m | [30,31,33] |

approximately at 1585 cm$^{-1}$ while in IC and IC, adsorbed eggshells were observed around 1637 cm$^{-1}$ [42]. The O = C-N and O = C-O-C vibration bands characteristic for IC were observed around 733 cm$^{-1}$ and 682 cm$^{-1}$, respectively. Skeletal vibrations, involving C–C stretching within the ring, appear in the 1500–1300 cm$^{-1}$ region [43]. In samples, ring vibrations were observed around 1490, 1300, 1120 and 1078 cm$^{-1}$. C-Br, C-Cl and C-F vibration bands are characteristic bands for DM, CM and IC, respectively. The C-Br vibration band is observed in the range of 700–500 cm$^{-1}$, while the C-Cl vibration band is observed in the region of 780–720 cm$^{-1}$ [40], and the C-F vibration band is observed in the region of 1400–900 cm$^{-1}$ [44]. In this study, C-Br, C-Cl, C-F vibration bands were observed around 711, 757 and 1016 cm$^{-1}$, respectively.

In order to observe the thickness of the eggshells, photographs were taken by placing the eggshells in a vertical position and are presented in figure 4.

When table 3 is examined, it is determined that the characteristic vibration bands of insecticides are more pronounced in *CCJ* eggshells after adsorption. In addition, it is seen in figure 4 that the thinnest eggshells are *CCJ*. This means thin eggshells absorb more pesticides than thicker ones.

## 3.3. UV-Vis spectroscopic analysis

An equilibrium study is important for us to have information about the adsorption process. This kind of examination is used to describe the interaction between the adsorbent and molecule. Also, the absorption and desorption capacity of the adsorbent have been determined [45]. Adsorption and desorption processes of CM, DM and IC were determined by UV-Vis analysis. Adsorption (%) and desorption (%) were obtained using equations (3.1) and (3.2).

$$Adsorption\ (\%) = \left( \frac{(C_i - C_e)}{C_i} \right) \times 100, \tag{3.1}$$

$$Desorption\ (\%) = \left( \frac{C_d}{C_a} \right) \times 100, \tag{3.2}$$

where $C_i$ and $C_e$ are insecticide concentration (mgL$^{-1}$) before and after adsorption, respectively. In addition, $C_d$ is the insecticide concentration after desorption; $C_a$ is the adsorbed insecticide concentration. Time-dependent changes of adsorption capacities for CM, DM and IC among these eggshells are given in figure 5. When the results were examined, it was seen that the *CCJ* had the highest adsorption capacity. According to figure 5, it was seen that most of the adsorption process was completed in the first 12 h in all samples, and at the end of 28 h, it was in the equilibrium state. The time-dependent removal of insecticide from the eggshells is given in figure 6. When figure 6 is examined, it is seen that IC desorption from eggshells is higher than others. In addition, it is seen that most of the desorption process is completed in the first 6 h in all eggshells.

The summary of figures 5 and 6 is given in table 4.

When table 4 is examined, it is determined that the rate of desorption is higher in thin eggshells. According to table 4, the adsorption capacities of eggshells are in the range of 35–86%, and their desorption capacities are in the range of 7–25%; high-adsorption–low-desorption capacity indicate that the eggshells can be used as a good adsorbent.

**Table 3.** Assignment of all IR spectra was given as wavenumbers (cm$^{-1}$) (Pure insecticide and adsorbed to eggshell samples) symmetric stretching γ: in-plane deformation vibration β: out-of-plane deformation vibration, τ: torsion, s: strong, m: medium, b: broad, w: weak, vw: very week, sh: shoulder, sb: strong broad.

| assignment | CM | | | | | | DM | | | | | | IC | | | | | |
|---|---|---|---|---|---|---|---|---|---|---|---|---|---|---|---|---|---|---|
| | CM | DH | LBC | ITW | AC | CCJ | DM | DH | LBC | ITW | AC | CCJ | IC | DH | LBC | ITW | AC | CCJ |
| v(CH) | 3029 m | 2982 w | 2960 w | 2961 w | 2960 w | 2960 w | 3019 m | 3076 w | 3076 w | 3076 w | 3075 w | 3073 w | | | | | | |
| v(CH) | | | | | | | 2964 w | 2959 w | 2965 w | 2963 w | 2967 w | 2963 w | | | | | | |
| v(CH) | 2928 s | 2931 w | 2932 w | 2829 w | 2929 w | 2930 w | 2923 m | 2829 w | 2930 w | 2930 w | 2933 w | 2931 w | | | | | | |
| v(CH) | 2875 w | 2876 sh | 2875 w | 2874 w | 2874 w | 2875 w | 2872 w | 2875 w | 2874w | 2875 w | 2875 w | 2874 w | | | | | | |
| v(NC) | 2044 w | 2136 sh | - | 2133 sh | 2140 sh | 2140 sh | 1939 w | 2138 w | - | 2138 sh | 2135 w | 2142 w | | | | | | |
| v(OC) | 1742 s | | 1742 m | 1737 sh | 1739 sh | 1742 w | 1745 w | 1740 w | 1740 w | 1741 w | 1739 sh | 1741 w | 1738 s | 1744 s | 1743 s | 1744 s | 1742 s | 1740 s |
| v(CC) | | | | | | | 1611 m | 1642 w | 1634 w | | 1639 sh | 1650 w | 1695 m | 1698 s | 1698 s | 1698 s | 1698 s | 1698 s |
| v(CC) | 1589 s | | 1587 w | | 1584 sh | 1587 w | 1592 w | 1587 w | 1586 sh | 1587 vw | 1587 sh | 1587 w | 1639 b | 1637 w | 1637 w | 1639 w | 1637 w | 1637 w |
| δ(HCC) | 1487 s | | 1488 w | | | 1488 w | 1495 m | 1487 vw | 1489 vw | 1489 vw | | 1487 w | 1506 m | 1507 w | 1508 w | 1507 w | 1508 w | 1509 w |
| δ(HCH) | 1455 m | | | | | | 1455 s | 1416 sb | 1416 sb | 1417 sh | 1418 sb | 1416 sb | 1439 sb | 1439 w | 1440 w | 1440 w | 1439 m | 1440 m |
| v(CC) | | | | | | | | | | | | | 1399 m | 1402 w | 1400 w | 1401 w | 1402 m | 1401 s |
| v(NC) | | | | | | | | | | | | | 1337 m | 1337 w | 1338 w | 1336 w | 1339 w | 1340 w |
| τ (HCCN) | | | | | | | 1291 w | | 1296 sh | 1297 vw | | 1295 vw | | | | | | |
| δ(HCC) | 1243 s | | 1247 w | 1247 sh | 1250 w | 1247 w | 1249 m | 1274 w | 1275 w | 1274 vw | 1276 sh | 1273 w | 1271 m | 1270 w | | | | |
| v(OC) | | | | | | | 1170 w | | | | | | 1245 w | | | | | |
| v(OC) | | | | | | | | 1165 w | 1165 w | 1166 vw | 1166 vw | 1165 w | | 1248 w | 1246 w | 1248 w | 1248 w | 1246 m |
| δ(HCC) | 1124 s | | 1127 w | 1128 w | 1128 sh | | 1120 m | 1128 w | 1129 w | 1129 vw | 1128 sh | 1129 w | 1179 m | 1176 w | 1177 w | 1175 w | 1178 w | 1179 m |
| v(NC) | | | | | | | | | | | | | 1100 w | 1103 w | 1102 w | 1102 w | 1102 w | 1102 w |
| δ(HCC) | 1078 m | 1083 vw | 1077 w | 1078 w | 1078 w | 1077 w | 1076 vw | 1077 w | 1077 w | 1077 w | 1078 w | 1077 w | 1074 w | 1072 w | 1072 w | 1072 w | 1072 w | 1072 w |
| τHCCC | | | | | | | | | | | | | 1041 m | 1040 w | 1039 w | 1040 m | 1040 m | 1039 m |
| v(FC) | 999 w | | 999 w | | | | | | | | | | 1016 w | 1016 w | 1016 w | 1016 w | 1017 w | 1017 w |
| v(CC) | | | | | | 999 vw | | | | | | | | | | | | |

(Continued.)

**Table 3.** (Continued.)

| assignment | CM | | | | | | DM | | | | | | IC | | | | | |
|---|---|---|---|---|---|---|---|---|---|---|---|---|---|---|---|---|---|---|
| | CM | DH | LBC | ITW | AC | CC | DM | DH | LBC | ITW | AC | CC | IC | DH | LBC | ITW | AC | CC |
| τ(H(C) | 921 w | | 919 w | 921 sh | | 912 w | 876 m | 874 s | 874 s | 874 s | 874 s | 874 s | 958 m | 958 s | 958 s | 958 s | 958 m | 958 m |
| δ(H(C) | 873 w | | 874 s | | | 874 m | | | | | | | 872 w | 873 s | 873 s | 873 s | 873 s | 873 m |
| ν(OC)+ ν(BrC) | | | | | | | 796 s | 799 w | 799 sh | 799 sh | 799 w | 799 w | | | | | | |
| ν(OC)+ ν(ClC) | 769 s | | 757 w | | 758 sh | 758 vw | | | | | | | | | | | | |
| τ(OC(O) | | | | | | | | | | | | | 733 w | 733 m | 733 m | 733 w | 733 w | 733 w |
| ν(BrC) | | | | | | | 691 s | 710 m | 711 s | 710 s | 712 m | 713 m | | | | | | |
| τ(ONN(C) | | | | | | | | | | | | | 697 w | 682 w | 682 m | 682 w | 683 w | 683 w |
| ν(OC) | 691 s | | 693 m | | | 693 m | | | | | | | | | | | | |
| δ(FOF(C) | | | | | | | | | | | | | 615 w | | 617 w | | | |

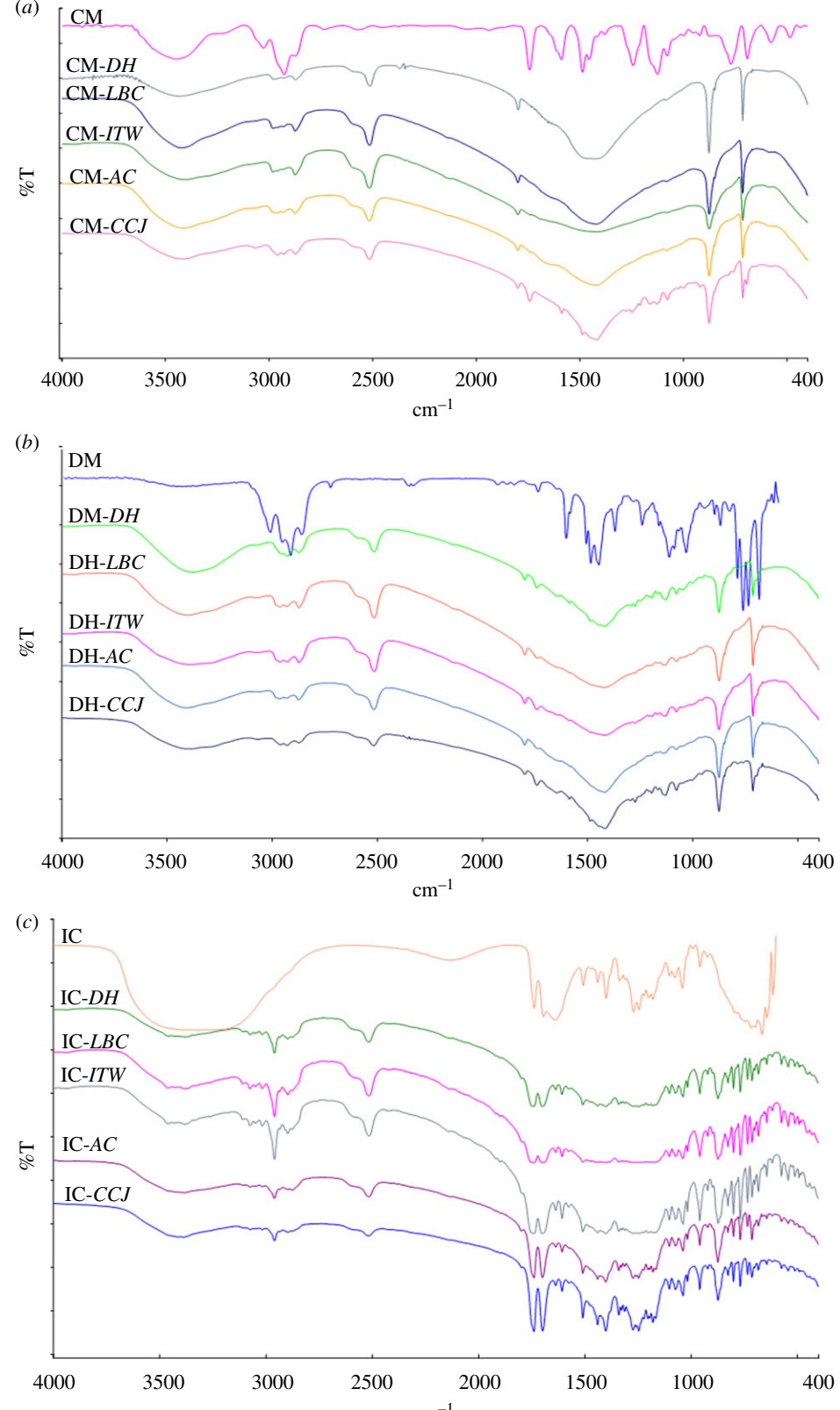

**Figure 3.** FTIR spectra of (*a*) CM and forms adsorbed on eggshells, (*b*) DM and forms adsorbed on eggshells and (*c*) IC and forms adsorbed on eggshells.

## 3.4. Theoretical analysis of electronegativity

HOMO-LUMO values of IC, CM and DM insecticides were calculated using the Gaussian 09 package program. The minimum energy required to remove an electron from the molecule is ionization energy

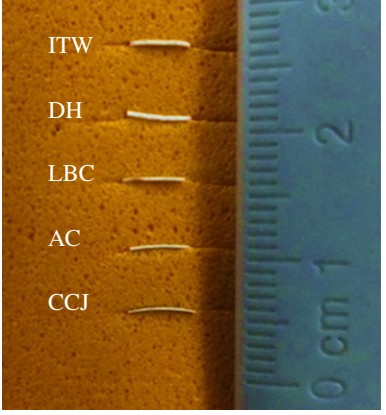

**Figure 4.** Thickness picture of eggshells.

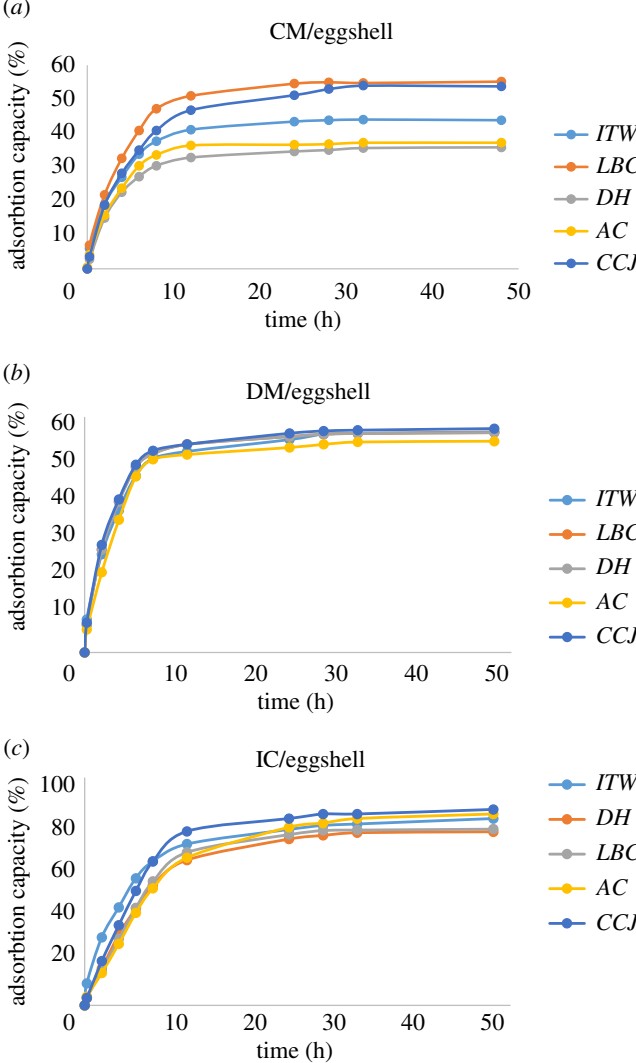

**Figure 5.** Effect of time on insecticide adsorption for (*a*) CM, (*b*) DM and (*c*) IC.

$I = -E_{\text{HOMO}}$ and the electron affinity, which is the energy amount that increases when an electron is added to the molecule. Its defined electronegativity parameter is $A = -E_{L\text{UMO}}$, $X = (I + A)/2$ which expresses the power to attract electrons of an atom in a molecule. Electronegativity values calculated for IC, CM and DM are presented in table 5.

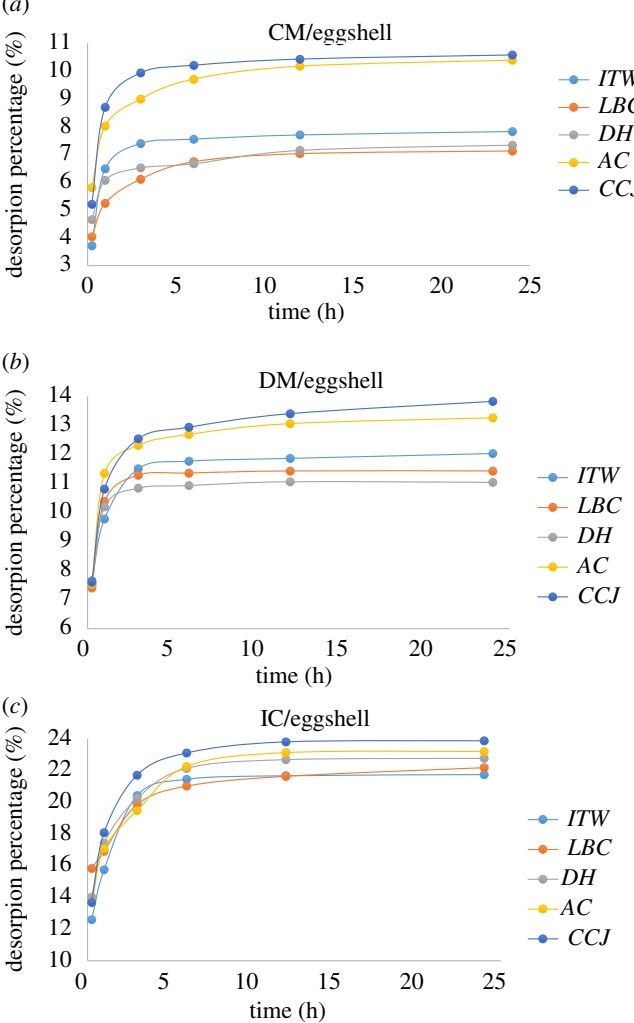

**Figure 6.** Effect of time on insecticides desorption for (*a*) CM, (*b*) DM and (*c*) IC.

**Table 4.** Adsorption and desorption results of insecticides to eggshell.

| insecticides | eggshell | adsorption (%) | desorption (%) |
|---|---|---|---|
| CM | *LBC* | 55,2 | 7,14 |
| | *CCJ* | 53,8 | 10,71 |
| | *ITW* | 43,8 | 7,85 |
| | *AC* | 37,2 | 10,47 |
| | *DH* | 35,8 | 7,34 |
| DM | *DH* | 60,4 | 11,12 |
| | *CCJ* | 57 | 13,92 |
| | *ITW* | 56,2 | 12,03 |
| | *LBC* | 56 | 11,53 |
| | *AC* | 53,8 | 13,36 |
| IC | *CCJ* | 86 | 23,93 |
| | *AC* | 84 | 23,34 |
| | *ITW* | 82 | 21,88 |
| | *LBC* | 77,4 | 22,49 |
| | *DH* | 76,2 | 22,83 |

**Table 5.** Calculated electronegativity values for IC, CM and DM.

| | $I = -E_{HOMO}$ (eV) | $A = -E_{LUMO}$ (eV) | electronegativity parameter $\left(\chi = (I + A)/2\right)$ (eV) |
|---|---|---|---|
| IC | 5,8156 | 3,1358 | 4,4757 |
| DM | 6,39522 | 1,6463 | 4,0207 |
| IC | 6,7544 | 2,4787 | 4,6165 |

# 4. Conclusion

Porous materials like eggshell have attracted much attention due to their unique structure and chemical properties. In this study, we aimed to remove insecticides from the environment by using some eggshells. Before and after adsorption were used to examine the surface structure of eggshells in the AFM images. The structure and chemical analysis of eggshells and also the adsorption process were investigated by spectroscopic analysis. And desorption of insecticides was determined by UV-Vis spectrophotometer. CM, DM and IC were characterized by FTIR spectroscopy. We compared the experimental results of our study before and after adsorption according to AFM and FTIR Spectroscopy. We used UV-Vis spectroscopy to detect adsorption and desorption capacity. According to the results of FTIR spectroscopy, the appearance of the characteristic vibration bands of insecticides in the samples after adsorption indicates that adsorption occurred on the eggshells. While C = N and C-O-C vibration bands are characteristic for CM and DM, O = C-O-C and O = C-N vibration bands are characteristic bands for IC. Characteristic peaks are concentrated in the range of 1742–1016 cm$^{-1}$ for IC. The observation of these vibrational bands in FTIR spectra can be regarded as evidence of the relationship between eggshells and insecticides. It is known that CaCO$_3$, which makes up more than 95% of eggshells, adsorbs high electronegativity solution [46]. When electronegativity values of insecticides are calculated by Gaussian 09, it is seen that IC has the highest electronegativity. Considering this property of IC, the adsorption mechanism can be explained via hydroxyl groups on eggshell. Aqueous solution of insecticides gives some electronical affinity for eggshell [47].

In addition to determining the ability of eggshells to adsorb insecticides, a desorption study was conducted to investigate the release of insecticides. Both studies were performed using the UV spectrophotometric analysis technique. It is seen that the adsorption capacity changes depending on the type of eggshells and insecticides. The highest adsorption and desorption capacity were measured as 86% and 23.93%, between *CCJ* eggshell and IC insecticide, respectively. Also, for all insecticides, the highest desorption rate occurred in *CCJ* eggshell. Low-desorption capacity was observed at thick eggshells like *LBC, DH* and *ITW*. It can be explained that molecules of insecticides could be trapped in the channels of eggshells during the adsorption phase. Hence, the release of insecticides from the structure of eggshell is low during the desorption process.

Time-dependent adsorption graphs show that most of the adsorption process was completed in 12 h for all samples and reaches equilibrium position at 28 h.

In the morphology of pure dried eggshells, there are many pores and pits distributed all over the surface randomly from one place to another. It has been evaluated from the two- and three-dimensional images of the eggshells that the surface of *CCJ* has more hills and pits than that of DH. According to the AFM images, the porous structure of the eggshells flattened after adsorption. The reason for this flattening is thought to be due to the filling of the pores in the eggshells by insecticides.

Changes in AFM images were observed as band shifts in FTIR and differences in absorbance values in UV-Vis as well. We observed from the FTIR and UV-Vis analytic results that IC is the best adsorbant among the insecticides used in this study. Also, as a second result, eggshells are a good adsorbent to remove insecticides. Eggshell is an easily accessible waste material. With this study, it was shown that waste eggshells can be used as an environmentally friendly adsorbent.

Ethics. Eggshells used in this study were obtained from the Isparta University of Applied Science of Agriculture Faculty Education, Research and Application Farm. Eggshells from chickens are used as waste materials on this farm. We obtained eggshells by asking the manager of the farm. They regarded the eggshells as waste materials which they had to remove from the farm. We used eggshells without any interaction (no touch) with the chickens.
Data accessibility. This article does not contain any additional data.
Authors' contributions. N.K.K. made the experimental and theoretical analysis of this study, interpreted the FTIR and UV-Vis analysis data, and drafted it. she contributed to the design and writing process of the article. B.T. designed,

coordinated this study and made a great contribution to writing the article. D.T.A. provided biomaterials, contributed to laboratory studies and researching the literature and gave support to write the article. All authors gave final approval for publication.

Competing interests. We declare we have no competing interests.

Funding. This research did not receive any specific grant from funding agencies in the public, commercial, or not-for-profit sectors.

Acknowledgement. The authors would like to thank Dr Öğr. Üyesi Gökhan Tüzün and Öğr. Gör. Veli Can Başkar for their contributions.

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
