## [Peer Review File · Royal Society Open Science]

Review History

RSOS-210100.R0 (Original submission)

Review form: Reviewer 1

Is the manuscript scientifically sound in its present form?

Yes

Are the interpretations and conclusions justified by the results?

Yes

Is the language acceptable?

Yes

Do you have any ethical concerns with this paper?

No

Have you any concerns about statistical analyses in this paper?

No

Recommendation?

Major revision is needed (please make suggestions in comments)

Comments to the Author(s)

The manuscript is well organized. There is some points to be clarified in the submitted manuscript.

The references which support FTIR assignment should be added in Table2.

The all original IR spectra in Table 3 should be given as Figure.

“time (h)” should be added in Figure 5.

“Referances” should be corrected as “References”.

Review form: Reviewer 2

Is the manuscript scientifically sound in its present form?

Yes

Are the interpretations and conclusions justified by the results?

Yes

Is the language acceptable?

Yes

Do you have any ethical concerns with this paper?

No

Have you any concerns about statistical analyses in this paper?

Yes

Recommendation?

Accept with minor revision (please list in comments)

Comments to the Author(s)

Authors have mentioned about the time-dependent changes of adsorption capacities in Fig 4 and about the time-dependent removal of insecticide in Fig5. How many times have they performed the experiment ? Is any kind of statistical analysis done?

Authors have claimed that Low desorption capacity was observed at thick eggshells like LBC any specific study conducted to understand this phenomenon kindly specify.

Decision letter (RSOS-210100.R0)

Dear Dr Kaya Kınaytürk:

Title: Eggshell as a biomaterial can have a sorption capability on its surface: A spectroscopic research

Manuscript ID: RSOS-210100

The editor assigned to your manuscript has now received comments from reviewers. We would like you to revise your paper in accordance with the referee and Subject Editor suggestions which can be found below (not including confidential reports to the Editor). Please note this decision does not guarantee eventual acceptance.

Please submit your revised paper before 07-May-2021. Please note that the revision deadline will expire at 00.00am on this date. If we do not hear from you within this time then it will be assumed that the paper has been withdrawn. In exceptional circumstances, extensions may be possible if agreed with the Editorial Office in advance. We do not allow multiple rounds of revision so we urge you to make every effort to fully address all of the comments at this stage. If deemed necessary by the Editors, your manuscript will be sent back to one or more of the original reviewers for assessment. If the original reviewers are not available we may invite new reviewers.

On behalf of the Subject Editor Professor Anthony Stace and the Associate Editor Dr Dattatray Late.

RSC Associate Editor:
Comments to the Author:

(There are no comments.)

RSC Subject Editor:

Comments to the Author:

(There are no comments.)

Reviewers' Comments to Author:

Reviewer: 1

Comments to the Author(s)

The manuscript is well organized. There is some points to be clarified in the submitted manuscript.

The references which support FTIR assignment should be added in Table2.

The all original IR spectra in Table 3 should be given as Figure.

"time (h)" should be added in Figure 5.

"Referances" should be corrected as "References".

Reviewer: 2

Comments to the Author(s)

Authors have mentioned about the time-dependent changes of adsorption capacities in Fig 4 and about the time-dependent removal of insecticide in Fig5. How many times have they performed the experiment? Is any kind of statistical analysis done?

Authors have claimed that Low desorption capacity was observed at thick eggshells like LBC any specific study conducted to understand this phenomenon kindly specify.

Author's Response to Decision Letter for (RSOS-210100.R0)

See Appendices A & B.

Decision letter (RSOS-210100.R1)

Dear Dr Kaya Kınaytürk:

Title: Eggshell as a biomaterial can have a sorption capability on its surface: A spectroscopic research

Manuscript ID: RSOS-210100.R1

It is a pleasure to accept your manuscript in its current form for publication in Royal Society Open Science. The chemistry content of Royal Society Open Science is published in collaboration with the Royal Society of Chemistry.

===COVID-SPECIFIC TEXT -- WILL ONLY BE ADDED TO COVID-PAPERS BY THE EDITORIAL OFFICE===

COVID-19 rapid publication process:

We are taking steps to expedite the publication of research relevant to the pandemic. If you wish, you can opt to have your paper published as soon as it is ready, rather than waiting for it to be published the scheduled Wednesday.

This means your paper will not be included in the weekly media round-up which the Society sends to journalists ahead of publication. However, it will still appear in the COVID-19 Publishing Collection which journalists will be directed to each week (<https://royalsocietypublishing.org/topic/special-collections/novel-coronavirus-outbreak>).

If you wish to have your paper considered for immediate publication, or to discuss further, please notify openscience_proofs@royalsociety.org and press@royalsociety.org when you respond to this email.

====END OF COVID-SPECIFIC TEXT -- WILL BE REMOVED AS NECESSARY BY THE EDITORIAL OFFICE====

Yours sincerely,
Dr Ellis Wilde on behalf of Dr Laura Smith
Publishing Editor, Journals

On behalf of the Subject Editor Professor Anthony Stace and the Associate Editor Dr Dattatray Late.

RSC Associate Editor
Comments to the Author:
Authors have revised the manuscript and now suitable for publication.

Reviewer(s)' Comments to Author:

Appendix A

Reviewer: 1

The manuscript is well organized. There is some points to be clarified in the submitted manuscript.

-The references which support FTIR assignment should be added in Table2.

References are added to Table 2 and the current version is submitted in the manuscript.

- The all original IR spectra in Table 3 should be given as Figure.

All original IR spectra in Table 3 are added to the manuscript as a figure. The caption of the IR spectra is defined as "Fig.3. FTIR spectra of (a) CM and forms adsorbed on eggshells (b) DM and forms adsorbed on eggshells (d) IC and forms adsorbed on eggshells".

Since a new figure is added, the labels of the figures in the study have changed. The updated manuscript with new figure tags has been loaded.

-“time (h)” should be added in Figure 5.

"Time (h)" is added to Figure 5.

-“Referances” should be corrected as “References”.

“Referances” is corrected as “References”

All corrections requested by reviewer 1 have been made and added to the manuscript.

Appendix B

Reviewer 2

Authors have mentioned about the time-dependent changes of adsorption capacities in Fig 4 and about the time-dependent removal of insecticide in Fig5. How many times have they performed the experiment? Is any kind of statistical analysis done?

We performed the experiment 5 times and the results were almost the same range. So, we chose the average one, and eliminated the other results. UV-Vis analysis was carried out with a measurement that is the average value of three times repeated of the relevant measurement.

Authors have claimed that Low desorption capacity was observed at thick eggshells like LBC any specific study conducted to understand this phenomenon kindly specify.

In this study, we primarily aimed to investigate the adsorption capacity of insecticides on different egg shells, not the effects of eggshell thickness on the adsorption capacity. The purpose of our desorption study was to determine the stability of the adsorption study.

In the adsorption process, the surface area of the adsorbent is directly proportional to the adsorption capacity. The porous structure of eggshells increases the surface area and thus increases the adsorption capacity. In the examples we are interested in, Low desorption capacity includes low adsorption capacity at the same sample. For this reason, it is difficult to make a general evaluation just by looking at low desorption capacity. Because there are many factors that affect the adsorption capacity and the thickness of the eggshells. Such as the environmental condition of chicken, age of chicken, nutritional diet, incubation period. Since it was not our primary goal to examine these parameters in our study, we did not control them. Therefore, this phenomenon is difficult to explain because it requires a large number of agreements between parameters.

If there is a narrowing of the pore on the eggshells towards the inside of the shell, we can say that the insecticide molecule is more easily trapped here.

However, in order to write this view as a general comment, we need some more analytical methods such as EDX, EDS, or micro-Raman spectroscopy at the points we will choose on the pores on the eggshell, then examine the molecule belonging to the pesticide at the exact pore point.

Our study consists of only 5 types of eggshells and we can say that 3 of them are thick.

LBC, DH, and ITW are thicker than CCJ and AC. Looking at the desorption percentages in Table 4, when you look at results as they are groups; thicker and thinner. The desorption rate is lower for the thick ones among 5 types of eggshells. For CM, there is a noticeable difference in desorption between thick and thin shells, whereas for IC, this is a slight difference. For this reason, a desorption mechanism in thick eggshells is a mechanism that belongs not only to desorption but also to adsorption. Reference 47 in the manuscript is a possible article on adsorption.

[47] E. Rápó, . L. E. Aradi, . Á. Szabó, . K. Posta, R. Szép and S. Tonk, "Adsorption of Remazol Brilliant Violet-5R Textile Dye from Aqueous Solutions by Using Eggshell Waste Biosorbent," Sci. Rep. 2020. 10 (8385): 1-12. (<https://doi.org/10.1038/s41598-020-65334-0>)

In addition, there are publications stating that eggshell thickness or pore structures will form a barrier. (Chen X, Li X, He Z, Hou Z, Xu G, Yang N, et al.) (2019) Comparative study of eggshell antibacterial effectivity in precocial and altricial birds using *Escherichia coli*. PLoS ONE 14(7): e0220054. <https://doi.org/10.1371/journal.pone.0220054>.

To grasp the idea about the adsorption some schematic illustration and SEM photographs will help us.

In the schematic representation, the pores on the outer surface called cuticle are in the form of plug, which leads to the idea of the pesticide molecule being trapped.

Grellet-Tinner, G., Fiorelli, L. E., & Salvador, R. B. (2012). Water Vapor Conductance of The Lower Cretaceous Dinosaurian Eggs From Sanagasta, La Rioja, Argentina: Paleobiological And Paleoecological Implications For South American Faveoloolithid And Megaloolithid Eggs. *Palaios*, 27(1), 35–47. Doi:10.2110/Palo.2011.P11-061r

Figure 3 - Lateral view of the eggshell (bar: 100μm).

LA SCALA JR, N et al . Pore Size Distribution in Chicken Eggs as Determined by Mercury Porosimetry. *Rev. Bras. Cienc. Avic.*, Campinas , v. 2, n. 2, p. 177-181, Aug. 2000. <https://doi.org/10.1590/S1516-635X2000000200007>

G.Kulshreshtha, A.Rodriguez-Navarro, E.Sanchez-Rodriguez, T.Diep, M.T.Hincke, Volume 97, Issue 4, 1 April 2018, Pages 1382-1390, Microbiology and food safety, Cuticle and pore plug properties in the table egg, doi.org/10.3382/ps/pex409

In the eggshell cuticle, the structure of the pore plug may be responsible for the adsorption mechanism in our study.

Figure 1. Microscopic Cross-Section of an Egg Shell (Lucore, 1994)